# Improving Out-of-Distribution Generalization in SAR Image Scene Classification with Limited Training Samples

**Zhe Chen** [1,2]**, Zhiquan Ding** [1]**, Xiaoling Zhang** [2,*]**, Xin Zhang** [1] **and Tianqi Qin** [1]

[1] Multisensor Intelligent Detection and Recognition Technologies R&D Center of CASC, Chengdu 610100, China; 19138952990@163.com (Z.C.); 13350314996@163.com (Z.D.); 17608097306@163.com (X.Z.); tianqiqin2008@163.com (T.Q.)
[2] School of Information and Communication Engineering, University of Electronic Science and Technology of China, Chengdu 610097, China
[*] Correspondence: xlzhang@uestc.edu.cn

**Abstract:** For practical maritime SAR image classification tasks with special imaging platforms, scenes to be classified are often different from those in the training sets. The quantity and diversity of the available training data can also be extremely limited. This problem of out-of-distribution (OOD) generalization with limited training samples leads to a sharp drop in the performance of conventional deep learning algorithms. In this paper, a knowledge-guided neural network (KGNN) model is proposed to overcome these challenges. By analyzing the saliency features of various maritime SAR scenes, universal knowledge in descriptive sentences is summarized. A feature integration strategy is designed to assign the descriptive knowledge to the ResNet-18 backbone. Both the individual semantic information and the inherent relations of the entities in SAR images are addressed. The experimental results show that our KGNN method outperforms conventional deep learning models in OOD scenarios with varying training sample sizes and achieves higher robustness in handling distributional shifts caused by weather conditions, terrain type, and sensor characteristics. In addition, the KGNN model converges within many fewer epochs during training. The performance improvement indicates that the KGNN model learns representations guided by beneficial properties for ODD generalization with limited training samples.

**Keywords:** knowledge-guided neural network (KGNN); OOD generalization; limited training samples; synthetic-aperture radar (SAR) image scene classification

## 1. Introduction

Synthetic-aperture radar (SAR) is a microwave imaging system with unique capabilities, being usable in all weathers, timeless, and capable of long-range observation [1]; it is thus increasingly required in many applications in the military and civil fields [2]. To utilize SAR images, the first task is to recognize the semantic category of the scene, so that a series of adaptive detailed computer vision procedures can be conducted, like those for object detection in different backgrounds. As a fundamental precondition for the advanced interpretation of SAR images, SAR image classification has become a significant task and has witnessed rapid development in recent decades.

In early years, researchers mainly focused on classic classification methods, with a feature extractor and a trainable classifier [3], the performance of which highly depend on the intra-class stability and inter-class discrimination power of extracted handcrafted features. Some classic low-level handcrafted features, e.g., color histograms, Gabor transform texture features, the gray-level co-occurrence matrix, scale-invariant feature transform (SIFT) [4], and histogram of oriented gradients (HOGs) [5], were studied and widely used in early scene classification tasks. Later, to achieve a more comprehensive discrimination power, the mid-level feature learning method was proposed, based on low-level feature representation through a coding form, such as bag of features [6] and sparse representation [7]. However, it

is still difficult for these methods to fully characterize the abundance of object singularities, especially in complex inhomogeneous scenes. In addition, without a feedback mechanism from classification to feature extraction, it is impossible to ensure that the features extracted are ideally suited for classification purposes.

In recent years, deep learning methods have achieved great success in SAR image classification [8]. A hierarchical structure of features can be learned by using convolutional layers and back-propagation [9,10]. The abundant deep feature statistics supports cutting-edge accuracy and better robustness in complex scene classification [11,12]. However, these modern data-driven deep learning methods are developed based on a fundamental assumption that the training and testing data are independent and identically distributed (the i.i.d. assumption), which is not true for some real applications like maritime SAR image classification. The testing data, i.e., the scenes to be classified, are often different from those in the training sets due to the changes in weather, sea state, geographic location, device, imaging mode, and other factors. Data distributional shifts render a sharp drop in the performances of the classic deep learning algorithms, creating the out-of-distribution (OOD) generalization problem [13]. Many studies [14–16] have shown that models optimized solely with training errors fail dramatically (sometimes even worse than a random guess) under strong distributional shifts. Moreover, for some special SAR imaging platforms, the available sample size is quite small, further exacerbating the difficulty of the OOD problem.

The reason for the OOD problem is that classic supervised learning methods greedily absorb all dependencies in training data to minimize the training errors, while not all dependencies remain in unseen testing distributions [13]. Therefore, the principle of OOD generalization with limited training samples is to add additional constraints that can reflect universal characteristics through different domains (including unseen domains). In this paper, a knowledge-guided neural network (KGNN) is proposed to deal with the OOD generalization problem with limited training samples. Some classic unsupervised model-based methods are employed to obtain saliency features that highlight the landscape segmentation in the scene, and knowledge indicating the characteristics or inherent information of entities in maritime scene classification tasks are summarized based on saliency analysis. To report the knowledge to the data-driven neural network, we design a feature integration strategy. Three saliency maps along with the original SAR image are input into separate branches of a ResNet-18 backbone to generate feature embeddings, then concatenated, and propagated together into the remaining four residual blocks of ResNet-18, addressing both the information of the pre-identification results and their inherent relations. The objective of KGNN is to boost the OOD generalization ability with very few available training samples, the quantity and diversity of which are limited by weather conditions, terrain type, and sensor characteristics. The major novelties of this work are as follows:

(1) We define knowledge as task-specific information about relations between entities in a maritime SAR image scene and extract the knowledge in descriptive sentence form through saliency analysis from the perspective of frequency and amplitude.

(2) We design a feature integration strategy to reflect the inherent information of objects (knowledge) in maritime scene classification tasks, and thus propose a new KGNN network.

The knowledge extracted via saliency analysis is universal through different domains, and can thus boost KGNN's OOD generalization ability. The experimental results demonstrate that the proposed KGNN surpasses advanced supervised learning methods with a state-of-the-art ODD generalization performance, especially with limited training data. The KGNN model converges within many fewer epochs during training, indicating that the parameter optimization is directed by knowledge, not solely based on gradients. The KGNN model also shows robustness in OOD scenarios affected by independent factors such as weather conditions, terrain type, and sensor characteristics, with varying degrees of improvements in the OOD generalization ability compared to the baseline methods. Improving the OOD generalization in SAR image scene classification with limited training samples can provide reliable scene classification results for new or niche radar detection

platforms, such as drone SAR imaging. In these systems, the image distribution often deviates from those of public datasets, and there is usually a scarcity in both the diversity and volume of accessible images.

## 2. Related Works

In the following, we review several bodies of literature that are relevant to the objective of our paper.

### 2.1. Unsupervised Representation Learning

OOD generalization is difficult since we have no access to samples in the test distribution, and if the test distribution is arbitrary or unrelated to the training distribution, the OOD generalization is then unsolvable [17]. Therefore, assumptions on how test distributions may change are necessary for OOD generalization. Some researchers have assumed that some properties of the training data describe spurious correlations and others may represent the phenomenon of interest, which is stable in unseen data. If these properties can be separated and identified, it can potentially benefit the OOD generalization [18–21]. These researchers proposed unsupervised representation learning methods based on this assumption, mainly including disentangled representation learning and causal representation learning. Dittadi et al. investigate how disentangled representations can be used for downstream tasks in different domains and scenarios [22]. Träuble et al. explored how the correlation between factors of variation in data affects the learning of disentangled representations [23]. Yang et al. propose a new framework for learning disentangled representations, that incorporates causal structure as a prior, which can learn more interpretable and transferable representations [20]. However, whether disentangled representation benefits OOD generalization remains controversial. Leeb et al. [24] conduct some quantitative extrapolation experiments, finding that the learned disentangled representation fails to extrapolate to unseen data. For causal representation, Träuble et al. found existing methods fail to capture the true causal structure of the data when there is correlation [23]. The challenge is, when the available amount of data is small, one cannot really tell causality from coincidence.

### 2.2. Supervised Model Learning

Other researchers have assumed that if the representations remain invariant when the domain varies, the representations are then transferable and robust on different domains, including unseen domains [25–28]. Supervised model learning methods are proposed based on this assumption, mainly including domain-adversarial learning and domain alignment. For domain-adversarial learning, Ganin et al. propose a domain-adversarial neural network (DANN) for domain adaptation. By adding a domain classifier that is trained in an adversarial manner, the DANN is trained to not only perform well on the main image classification task, but also to adapt to the domain shift between training data and real-world data [29]. Gwon et al. introduced a new adversarial mixup (AM) training method, which generates OOD samples that significantly diverge from the support of the training data distribution but are not completely disjoint. The OOD samples are used to synthesize differently distributed data for training to increase the OOD generalization ability [30]. For domain alignment, the domain-invariant representations are learned via the alignment of features [27,31]. Motiian et al. propose learning semantic alignment between different domains by minimizing the distance between samples from different domains but the same class, and maximizing the distance between samples from different domains and classes [28]. Shao et al. propose a new framework that uses a multi-adversarial strategy to align the feature distributions of different domains. It can generalize to unseen face presentation attacks by learning a shared and discriminative feature space from multiple source domains [26]. However, these supervised model learning methods need a fundamental diversity in the domain sources to learn the domain-invariant features, which cannot be satisfied in the OOD case with few training samples available.

*2.3. Few-Shot Learning*

There are also some studies that focus on metric-based few-shot learning for SAR ship classification. These studies usually employ a Siamese network structure and a triplet loss, with some additional techniques to further increase the robustness and accuracy of classification, such as a dense connected convolutional network [32] and feature fusion [33]. The basic idea for metric-based few-shot learning is to train the model with a large amount of data samples on multiple categories, and during testing, the model is provided with novel categories (also referred to as a novel set) where there are multiple data samples, usually with few data shifts for each category [34]. One testing sample is chosen to be fed into the template Siamese branch and other testing samples are fed into other Siamese branches to determine whether they are a "match", thus giving the classification results. The task of few-shot learning is different from the task discussed in this paper: 1. the quantity and diversity of the available trainable data in our task are highly limited. 2. The method of template matching via a Siamese network may be impractical due to the vast diversity of testing data within the same category.

*2.4. Integrating Knowledge into Deep Learning*

For OOD cases with limited training samples, additional constraints are required to achieve reliable predictions. Humans are often able to learn without direct examples, opting instead for high-level instructions or guidelines for how a task should be performed [35], which means a high-level extrapolation ability. If the properties that can benefit OOD generalization are already known, it is more effective to assign this knowledge to the model, rather than letting the model learn by itself. Therefore, many studies extract these constraints from prior domain knowledge, e.g., from known laws of physics or expert experience, focusing on how to improve machine learning models by additionally incorporating prior knowledge into the learning process [36]. Stewart et al. integrated a parabola function into the loss function, and successfully tracked the height of balls in free fall without providing labels [35]. Diligenti et al. used semantic-based regularization (SBR) as the underlying framework to represent the prior knowledge as seen in images [37]. In SAR classification applications, Huang et al. proposed a physics-guided and -injected learning (PGIL) model, which employs a physics-guided network to convert prior knowledge as feature embeddings, then employs a physics-injected network to introduce the physics-aware features into a CNN pipeline [38]. Zhang et al. preliminarily explored the possibility of the injection of traditional handcrafted features into modern CNN-based models to further improve SAR ship classification accuracy [39]. The knowledge-integrated deep learning methods tend to outperform the pure data-driven methods, strengthening the interpretability and physics consistency of the predictions. However, though theoretically feasible, few of these methods have been used in solving the OOD generalization problem with limited data.

## 3. Materials and Methods

For the practical application of maritime SAR image scene classification, it is quite common that scenes encountered in actual tasks are different from those in pre-training due to the weather, sea state, geographic location, device, imaging mode, motion errors, and other factors. Figure 1 shows the distributional shift between different series of SAR images divided by imaging time, location, mode, and device. It can be observed that the features of image components, such as sea, land facilities, and landscapes, vary greatly between series. Additionally, for some special SAR imaging platforms, the number of available images can be quite small. Our goal is to identify if a scene contains land or port facilities (bridge, oil tanks, harbor, etc.) by classifying a vast number of images from unseen series, using only a small selection of images from a few series for training.

**Table 1.** Detailed information of MSAR 1.0 dataset. For multiple-factor influence experiments, the yellow shading outlines the series we chose to establish as the training set, the rest of the series were used to establish the testing set.

| Series | Image Index | Time | Location | Satellite | Imaging Mode |
|---|---|---|---|---|---|
| 1 | 1~496 | 24 Mar. 2021 | - | HISEA-1 | SM |
| 2 | 613~5251 | 15 Jan. 2017 | E122.0, N30.3 | Gaofen-3 | FSI |
| | 5252~5351 | 24 Oct. 2017 | E120.8, N36.1 | Gaofen-3 | FSI |
| | 5352~5745 | 24 Oct. 2017 | E120.9, N35.7 | Gaofen-3 | FSI |
| 3 | 5746~5884 | 24 Oct. 2017 | E121.0, N35.2 | Gaofen-3 | FSI |
| | 5885~5936 | 24 Oct. 2017 | E121.1, N34.7 | Gaofen-3 | FSI |
| | 5937~8229 | 24 Oct. 2017 | E122.0, N30.1 | Gaofen-3 | FSI |
| 4 | 8230~8243 | 16 Nov. 2017 | E110.5, N18.1 | Gaofen-3 | NSC |
| 5 | 8244~8804 | 5 Jul. 2017 | E120.1, N35.8 | Gaofen-3 | QPSI |
| 6 | 8805~8913 | 12 Aug. 2017 | E109.7, N18.4 | Gaofen-3 | UFS |
| 7 | 8914~10,119 | 15 Jul. 2017 | E120.4, N35.4 | Gaofen-3 | FSII |
| 8 | 10,120~12,972 | 3 Nov. 2017 | E121.9, N30.1 | Gaofen-3 | QPSI |
| 9 | 12,973~12,974 | 20 Feb. 2017 | E120.7, N35.0 | Gaofen-3 | FSI |
| | 12,975~13,273 | 20 Feb. 2017 | E120.9, N36.0 | Gaofen-3 | FSI |
| 10 | 13,274~13,635 | 6 Jul. 2017 | E129.6, N33.0 | Gaofen-3 | QPSI |
| | 13,636~14,145 | 6 Jul. 2017 | E129.7, N33.5 | Gaofen-3 | QPSI |
| | 14,146~14,314 | 2 Sept. 2017 | E129.6, N33.1 | Gaofen-3 | QPSI |
| 11 | 14,315~14,387 | 2 Sept. 2017 | E129.7, N33.4 | Gaofen-3 | QPSI |
| | 14,388~14,411 | 2 Sept. 2017 | E129.7, N33.6 | Gaofen-3 | QPSI |
| 12 | 14,412~15,432 | 30 Sept. 2017 | E120.5, N36.3 | Gaofen-3 | FSII |
| | 15,433~17,588 | 30 Sept. 2017 | E121.9, N30.3 | Gaofen-3 | FSII |
| 13 | 17,589~19,121 | 15 Feb. 2017 | E122.3, N29.9 | Gaofen-3 | FSI |
| 14 | 19,122~21,147 | 10 Jul. 2017 | E122.5, N30.2 | Gaofen-3 | NSC |
| 15 | 21,148~22,875 | 29 Jul. 2017 | E121.1, N30.5 | Gaofen-3 | NSC |
| | 22,876~23,202 | 5 Oct. 2017 | E120.4, N36.2 | Gaofen-3 | QPSI |
| | 23,203~23,264 | 5 Oct. 2017 | E121.0, N33.5 | Gaofen-3 | QPSI |
| | 23,265~24,122 | 5 Oct. 2017 | E121.9, N30.1 | Gaofen-3 | QPSI |
| | 24,123~24,147 | 5 Oct. 2017 | E121.5, N34.2 | Gaofen-3 | QPSI |
| 16 | 24,148~24,157 | 5 Oct. 2017 | E121.6, N34.7 | Gaofen-3 | QPSI |
| | 24,158~24,168 | 5 Oct. 2017 | E121.7, N35.0 | Gaofen-3 | QPSI |
| | 24,169~24,187 | 5 Oct. 2017 | E121.8, N35.6 | Gaofen-3 | QPSI |
| | 24,188~24,217 | 5 Oct. 2017 | E122.1, N36.7 | Gaofen-3 | QPSI |
| 17 | 24,218~24,229 | 15 Oct. 2017 | E124.6, N34.7 | Gaofen-3 | QPSI |
| | 24,230~24,248 | 15 Oct. 2017 | E124.7, N35.2 | Gaofen-3 | QPSI |
| | 24,249~24,263 | 3 Nov. 2017 | E121.0, N35.0 | Gaofen-3 | QPSI |
| | 24,264~24,327 | 3 Nov. 2017 | E121.1, N35.4 | Gaofen-3 | QPSI |
| 18 | 24,328~24,378 | 3 Nov. 2017 | E121.1, N35.6 | Gaofen-3 | QPSI |
| | 24,379~24,396 | 3 Nov. 2017 | E121.2, N35.8 | Gaofen-3 | QPSI |
| | 24,397~24,429 | 3 Nov. 2017 | E121.3, N36.2 | Gaofen-3 | QPSI |
| | 24,430~24,435 | 3 Nov. 2017 | E121.4, N36.6 | Gaofen-3 | QPSI |
| | 24,436~25,080 | 15 Nov. 2017 | E122.7, N36.5 | Gaofen-3 | UFS |
| 19 | 25,081~25,461 | 15 Nov. 2017 | E123.0, N34.8 | Gaofen-3 | UFS |
| | 25,462~25,623 | 15 Nov. 2017 | E123.0, N35.0 | Gaofen-3 | UFS |
| | 25,624~25,675 | 15 Nov. 2017 | E123.1, N34.5 | Gaofen-3 | UFS |
| 20 | 25,676~26,892 | 6 Jan. 2017 | E132.5, N32.5 | Gaofen-3 | WSC |
| 21 | 26,893~28,449 | 4 Aug. 2017 | E128.9, N32.2 | Gaofen-3 | WSC |

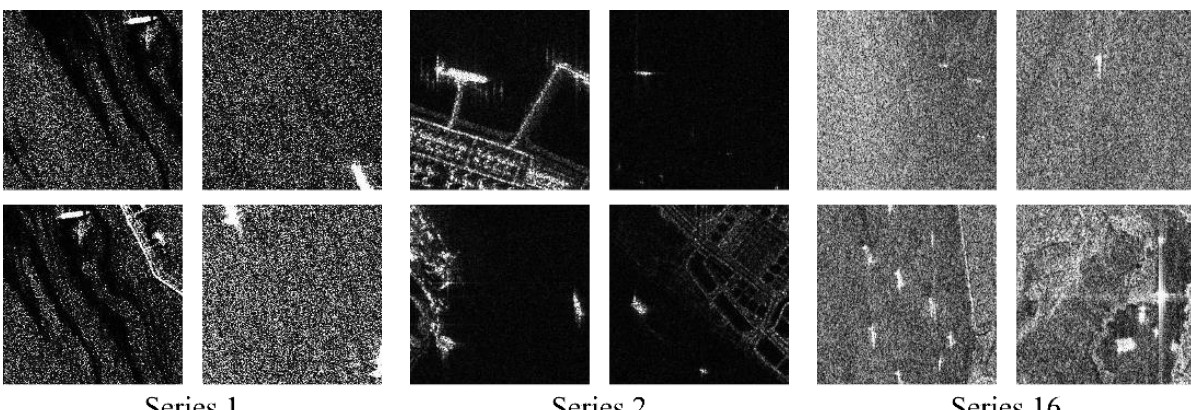

**Figure 1.** Three different series of SAR images with different imaging times, locations, modes, and devices (detailed information is shown in Table 1). Distinct characteristics in style and layout can be seen between these series, indicating distributional shifts.

We propose a KGNN model to overcome the OOD problem with limited training samples mentioned above by additionally incorporating prior knowledge into the deep learning process. Intuitively, the advanced CNN-based model is still regarded as the main body of the classifier, because its classification performance is commonly better than traditional ones. The meaning of knowledge is difficult to define in general and is an ongoing debate in philosophy [36]. In this work, we assume knowledge as task-specific information about relations between entities in maritime SAR image scenes. We extract the knowledge in descriptive sentence form through saliency analysis in Section 3.1 and propose a KGNN model to incorporate the knowledge into a deep learning backbone in Section 3.2.

### 3.1. Knowledge Extraction via Saliency Analysis

Figure 1 shows that though the characteristics of entities such as sea, land structures, and landscapes differ significantly across series, the distinctions between these entities, such as between the sea and land/port facilities or the sea and ships, remain relatively consistent in every image series. Based on this observation, we assume a domain-invariant pre-identification of land regions, marine regions, and targets may be a boost and a good initialization for OOD generalization. Therefore, we first employ the saliency features proposed in our previous study [40] to pre-identify the land and marine regions. Saliency refers to the contrast of an item from its surroundings. For the segmentation task in maritime SAR images, the main challenge is the high confusion between landforms and sea clutter under speckle noise. In such circumstances, a bottom-up region-merging technique such as multiresolution segmentation [41] would be quite time consuming and inaccurate due to speckle noise [40]. To address the scale difference between noise, sea clutter, and landforms, a second-order Gaussian regression filter [42,43] is applied to highlight the low-frequency landforms. The filtration process can be defined by the following minimization problem [43]:

$$\int_0^{ly} \int_0^{lx} \rho \begin{pmatrix} Z_0(\xi,\eta) - Z_f(x,y) \\ -B_{10}(x,y)(\xi-x) - B_{01}(x,y)(\eta-y) \\ -B_{20}(x,y)(\xi-x)^2 - B_{02}(x,y)(\eta-y)^2 \\ -B_{11}(x,y)(\xi-x)(\eta-y) \end{pmatrix} S(\xi-x,\eta-y)d\xi d\eta \tag{1}$$
$$\Rightarrow min_{Z_f(x,y),B_{10}(x,y),B_{01}(x,y),B_{20}(x,y),B_{02}(x,y),B_{11}(x,y)}$$

where $Z_0$ is the input SAR image, $Z_f$ is the filtration result of $Z_0$ by zeroing the partial derivatives in the directions of $Z_f$, $B_{10}$, $B_{01}$, $B_{11}$, $B_{20}$, and $B_{02}$. $B_{10}(x,y)$ and $B_{01}(x,y)$ are first-order coefficients. $B_{20}(x,y)$, $B_{11}(x,y)$, and $B_{02}(x,y)$ are second-order coefficients. $\rho(r) = r^2/2$ is the error metric function of the estimated residual. $S(x,y) = \frac{1}{\alpha^2 \lambda_{cx} \lambda_{cy}} exp$

$\left[-\pi\left(\frac{x}{\alpha\lambda_{cx}}\right)^2 - \pi\left(\frac{y}{\alpha\lambda_{cy}}\right)^2\right]$ is the Gaussian weighting function, where $\alpha = \sqrt{log(2)/\pi}$. $\lambda_{cx}$ and $\lambda_{cy}$ are the cutoff wavelengths in the $x$ and $y$ directions, respectively. In our case, for an image with a size of $256 \times 256$ pixels, the cutoff wavelength is chosen as 32 pixels in both the x and y directions. Figure 2a shows an SAR image with a harbor scene, and Figure 2b shows its filtration result; the speckle noise is alleviated, and low-frequency landforms and targets are emphasized.

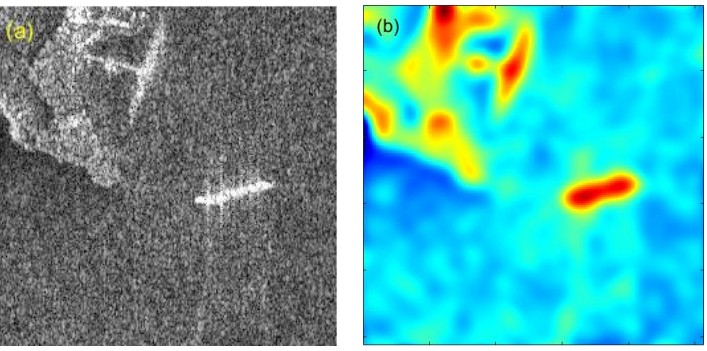

**Figure 2.** (**a**) Harbor scene satellite SAR image, and (**b**) the filtration result.

The filtration result is then rescaled to an intensity level of $[0, 255]$. For intensity $i \in \{0, 1, \ldots, 255\}$, the corresponding pixel number with intensity $i$ is $n_i$, and the probability of occurrence is $p_i = n_i / \sum_{i=0}^{255} n_i$. A multi-level Otsu's method that uses the maximized inter-class variance $f$ of the amplitude as the evaluation function for adaptive threshold selection is used [44]:

$$f = \frac{(\mu w_{t_1} - \mu_{t_1})^2}{w_{t_1}} + \cdots + \frac{\left(\mu_{t_j} - \mu w_{t_j} + \mu w_{t_{j-1}} - \mu_{t_{j-1}}\right)^2}{w_{t_j} - w_{t_{j-1}}} + \cdots + \frac{(\mu w_{t_n} - \mu_{t_n})^2}{1 - w_{t_n}} \quad (2)$$

where $w_{t_j}$ represents the cumulative probability of occurrence of gray-level interval $[0, t_j]$, and $w_{t_j} = \sum_{i=0}^{t_j} p_i$. $\mu_{t_j}$ denotes the mean value at gray-level interval $[0, t_j]$, and $\mu_{t_j} = \sum_{i=0}^{t_j} i \cdot p_i$. $\mu$ is the mean value of the filtration result and $\mu = \sum_{i=0}^{255} i \cdot p_i$. The thresholds $t = [t_1, t_2, \ldots, t_n]$ can be determined using the Nelder–Mead simplex method [45], in our case we use three thresholds and thus achieve a 4-level segmentation. The segmentation result of Figure 2a is shown in Figure 3a. By applying further morphological operations, two frequency saliency maps that outline the land (SM1) and marine regions (SM2) are obtained, as shown in Figures 3b and 3c, respectively.

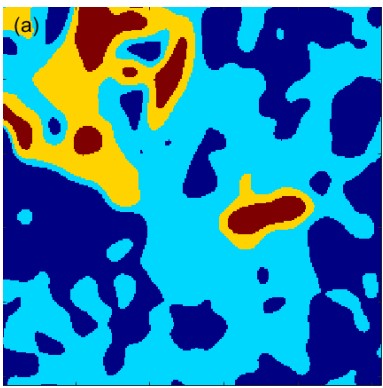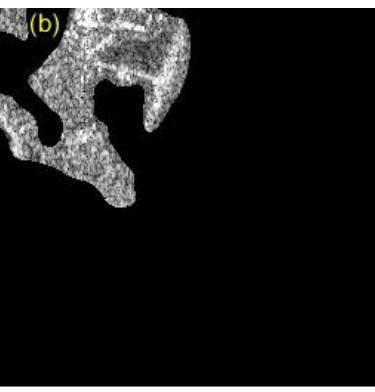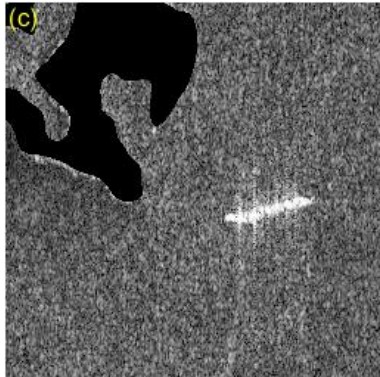

**Figure 3.** (**a**) A 4-level segmentation result of harbor scene; (**b**) frequency saliency map outlining land regions (SM1); (**c**) frequency saliency map outlining marine regions (SM2).

The distinction between targets and background clutter is another important domain-invariant feature which can be addressed by target detection. Pixel intensity is a fundamental feature of SAR images that can be used to detect targets. However, for inhomogeneous scenes, detection methods using a global intensity threshold would not be suitable since many false alarms such as strong sea clutter, noise, land facilities, and land clutter will also be detected [46]. In this paper, we adopt a local constant false alarm rate (CFAR) detector to estimate an adaptive intensity threshold for each pixel based on the statistical distribution of the background clutter in its vicinity. Figure 4a illustrates the local CFAR procedure: a small rectangle represents the protection window that excludes the internal pixels from the background estimation. A larger rectangle represents the background window, the pixels between the two windows are used to fit a statistical model and a threshold is thus derived. This threshold is then applied to examine whether the center pixel belongs to a target or not. In our case, we use a Gaussian distribution to model the background clutter, the adaptive threshold matrix $T_c$ can then be calculated by [47]

$$T_c = \Sigma \phi^{-1}(1 - pfa) + M \tag{3}$$

where $\Sigma$ and $M$ are the matrix of the standard deviation and mean of the pixel intensity values between the concentric windows, and $pfa$ is the false alarm rate (in our case 0.001). $\phi$ is the standard normal cumulative distribution function:

$$\phi(x) = \frac{1}{2}\left(1 - \text{erf}\left(-\frac{x}{\sqrt{2}}\right)\right) \tag{4}$$

$$\text{erf}(x) = \frac{2}{\sqrt{\pi}} \int_0^x e^{-t^2} dt \tag{5}$$

The matrix of the mean values $M$ can be obtained through mean filtering with kernel $K$. Kernel $K$ is a matrix the same size as the background window, with values of 0 for the area with the same size as the protection window in the center and 1 for the surrounding area, marked by dashed shading, as shown in Figure 4a. The matrix of standard deviation $\Sigma$ can be obtained by

$$\Sigma = \sqrt{K * Z_0{}^2 - (M)^2} \tag{6}$$

Figure 4b shows the matrix of adaptive threshold $T_c$, which is high over the land region, and low over the target region. The input image $Z_0$ is compared with $T_c$, and the part exceeding the threshold is the detection result. Figure 4c shows the result of the local CFAR detection. With an adaptive threshold, the target is successfully detected with very few false alarms. This detection forms our target saliency map (SM3).

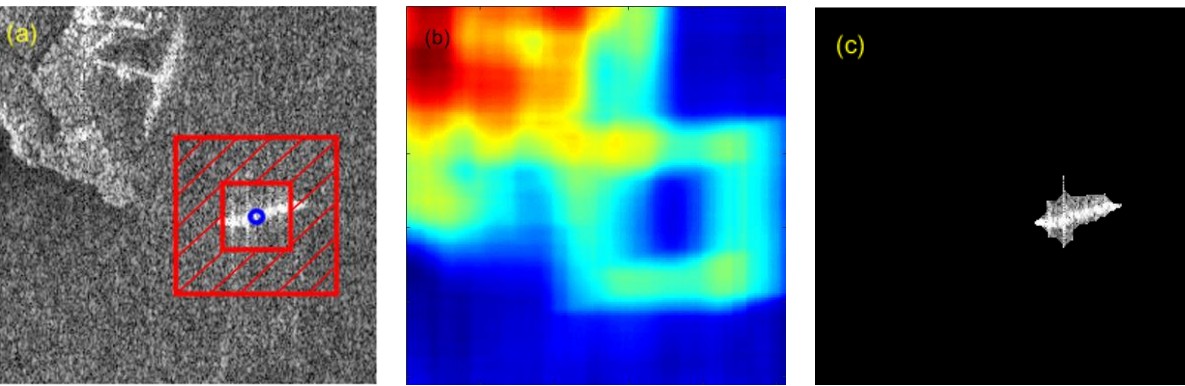

**Figure 4.** (**a**) Schematic of the local CFAR process; (**b**) the matrix of adaptive threshold $T_c$; (**c**) the local CFAR detection result (SM3).

To demonstrate the identification ability of these saliency features, pre-identification results of examples from various scenes are presented in Figure 5. For clean, weak sea clutter with multiple targets, as shown in Figure 5a, no entities are misidentified in SM1, and all targets are successfully identified in SM3. Figure 5b,c illustrate that as the amount of sea clutter increases, it is initially detected in part by SM1 (Figure 5b). Eventually, all sea clutter is recognized in SM1 while no entities are detected in SM2. Despite the strong sea clutter, the ship target can still be accurately identified in SM3 (Figure 5c). Figure 5d shows a ship target with energy leakage; the leakage is identified in SM1, while sea clutter is identified in SM2. Figure 5e shows inhomogeneous sea clutter, the stronger sea clutter is identified in SM1, the weaker sea clutter is identified in SM2, while the ship target among strong sea clutter is successfully identified in SM3. Figure 5f shows scenes with small amounts of dim land clutter; the land region is successfully identified in SM1, the sea region is identified in SM2, and all targets are identified in SM3. Figure 5g shows an embankment occupying a very small area in the image; the embankment can be successfully identified in SM1. Figure 5h shows dim land with a bright bridge; the land region is successfully identified in SM1, while the bridge is misidentified at the edge of land in SM3. Figure 5i shows land with oil tanks; the land region is successfully identified in SM1, while some small bright land facilities at the edge of land regions are misidentified in SM3. From Figure 5j–n, the distinction between sea and land becomes less pronounced, yet the identification remains accurate in SM1, SM2, and SM3 up to Figure 5n. Some strong sea clutter is identified alongside land clutter in SM1, as shown in Figure 5n.

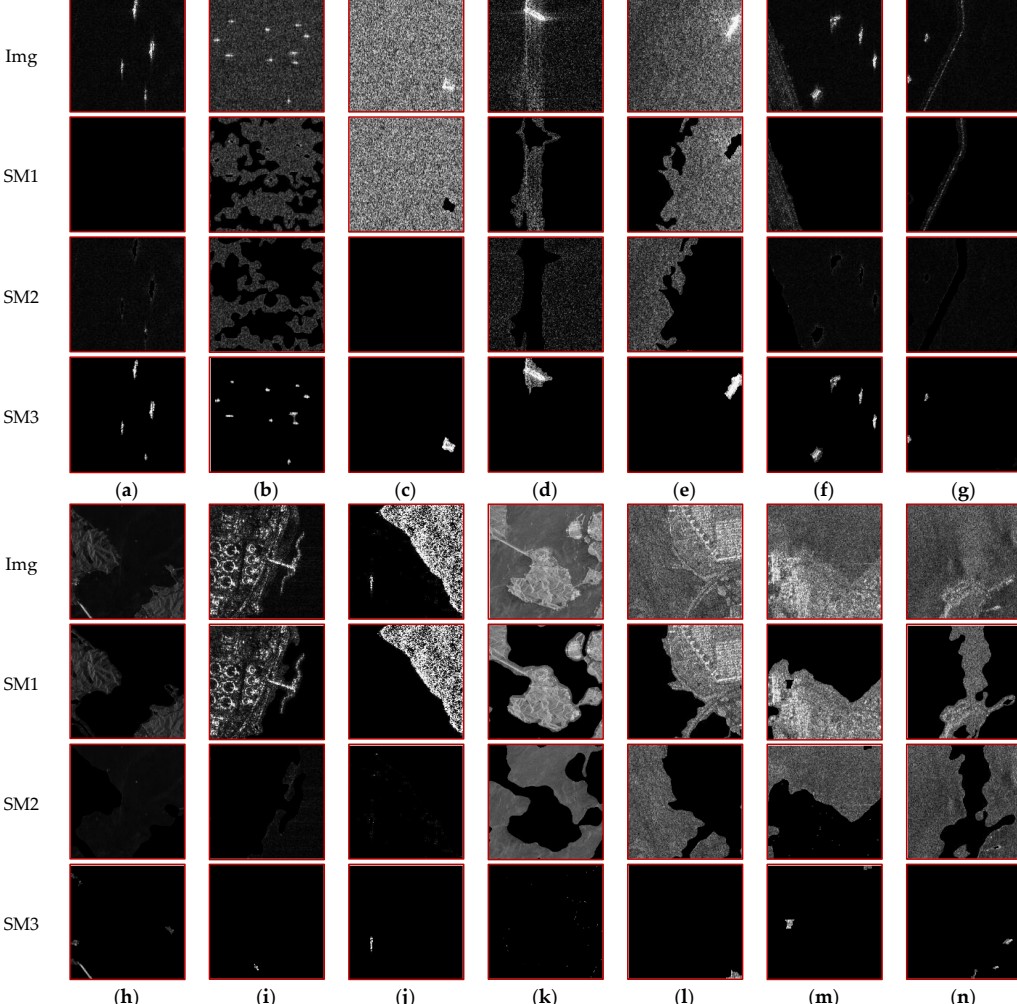

**Figure 5.** (**a**–**n**) Pre-identification results of SM1, SM2, and SM3 in various scenes.

It can be seen from the results that the pre-identification ability of the three saliency maps is quite good for scenes with good sea–land contrast or with clean, weak sea clutter. The pre-identification ability of SM1 and SM2 deteriorates for scenes with strong/inhomogeneous sea clutter, or with very confusing sea–land boundaries. For SM3, entities such as bright bridges and small bright facilities at the edge of land regions are often detected as targets rather than a portion of land. It is obvious that these pre-identification results cannot be simply input into the CNN networks since the error in each saliency map will also propagate and finally lead to false classification. However, by observing the pre-identification results of various scenes, we found that no matter whether they were good or deteriorated, there were certain consistent patterns that the pre-identification results of the saliency maps adhered to:

(1) Ship targets tend to appear in the center of strong sea clutter and at the edge of land, but rarely in the center of land.
(2) There is little difference in the uniformity of strong and weak sea clutter, while a significant difference is observed in the uniformity of sea and land clutter.
(3) The layout of land regions is more regular compared to strong sea clutter.

These laws reflect the relations between entities in maritime SAR image scenes based on the saliency features. Therefore, with an appropriate integration strategy, addressing the correlations between saliency features, these pre-identification results can be a boost and supplement for a data-driven deep learning model. The imperfections in the pre-identification results can be corrected by their inherent relations.

### 3.2. Knowledge-Guided Neural Network (KGNN)

SM1, SM2, and SM3 provide some very fundamental information about the layouts of the entities in an SAR image, with imperfections in several kinds of scene. We summarize some knowledge based on the saliency analysis and believe this knowledge can be a boost to OOD generalization with very limited training samples. According to the prior knowledge in Section 3.1, the inherent relations between the saliency maps need to be addressed. The sentence-form knowledge in qualitative descriptions cannot be incorporated into a deep learning model directly; therefore, the following integration strategy is designed:

(1) We feed the three saliency maps along with the original SAR image into separate branches of a ResNet-18 [48] backbone to generate feature embeddings. We apply the convolutional block and the first four residual blocks of ResNet-18 as the data-driven feature extractor branch. The purpose is to extract features with semantic discrimination beneficial to the classification of each saliency map. The original SAR image feature extractor branch is designed to address the neighboring information in the edge between saliency maps. In practice, we implement the four feature extractor branches using grouped convolutional layers [49] and grouped batch normalization layers [50].
(2) The feature embeddings generated from the four branches are concatenated and propagate together into the remaining four residual blocks of ResNet-18. The transformed semantic features of the saliency maps correlate with each other and evolute in the mid-level and high-level layers of the conventional classification network successively.

To avoid the loss of interactive information, the original image propagates in the same way with other saliency feature maps. Figure 6 depicts the idea of our KGNN model: 1. extract saliency features; 2. summarize knowledge using saliency analysis; 3. design integration strategy based on this knowledge. The KGNN model is thus established, the final output is the scene type of the input image. Figure 7 shows the detailed neural network architecture diagram of the KGNN model.

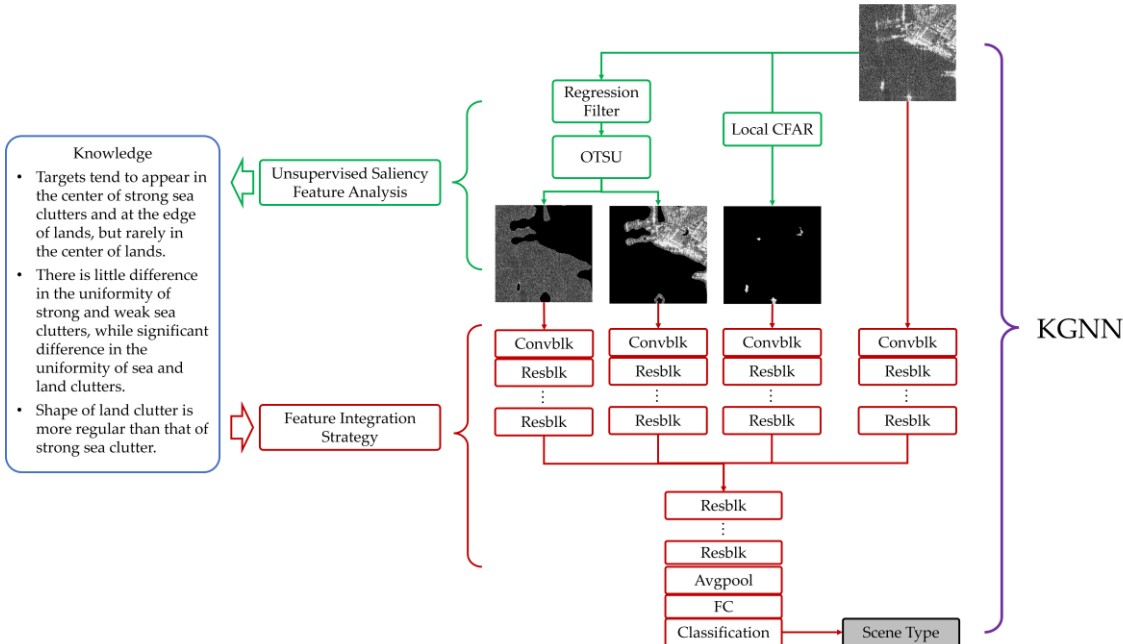

**Figure 6.** The idea of the proposed KGNN. Convblk stands for convolutional block; Resblk stands for the residual block in ResNet-18; Avgpool stands for average pooling; FC stands for fully connected layer.

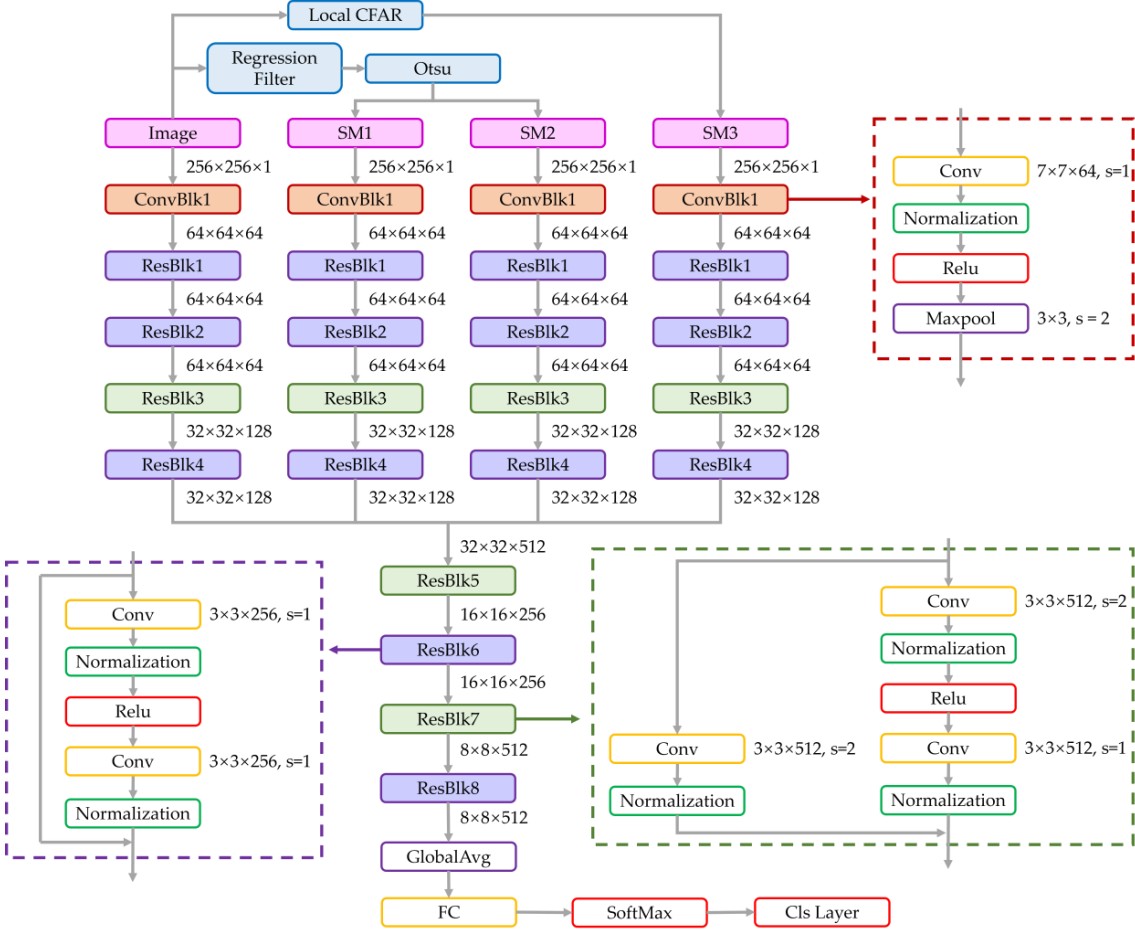

**Figure 7.** Architecture diagram of the KGNN.

## 4. Results and Discussion

### 4.1. Data Description

In this section, we evaluate our proposed method on the MSAR 1.0 dataset [51,52], which is a large-scale synthetic-aperture radar (SAR) image dataset for target detection. The dataset consists of 28,449 detection slices, each with a size of 256 × 256 pixels, collected from two moving satellite platforms: HISEA-1 and Gaofen-3. The dataset covers various scenarios such as airports, ports, inshore, islands, offshore, urban areas, etc., and includes four classes of targets: aircraft, oil tank, bridge, and ship (moving target). The HISEA-1 satellite has three imaging modes with resolutions ranging from 1 m to 10 m, the incident angle from 20° to 35°, and a single VV polarization. The Gaofen-3 satellite has 12 imaging modes with resolutions ranging from 1 m to 100 m, and four polarization modes, including HH, HV, VH, and VV. We list the detailed information of the MSAR dataset in Table 1. There are three key characteristics: the imaging time, which means different weather conditions; the imaging location, which corresponds to different scenarios; and the imaging modes, which affect the incident angle, resolution, and polarization. We divide the MSAR dataset into 21 series based on these three key characteristics. The style and layout of images can vary greatly across series, as shown in Figure 8.

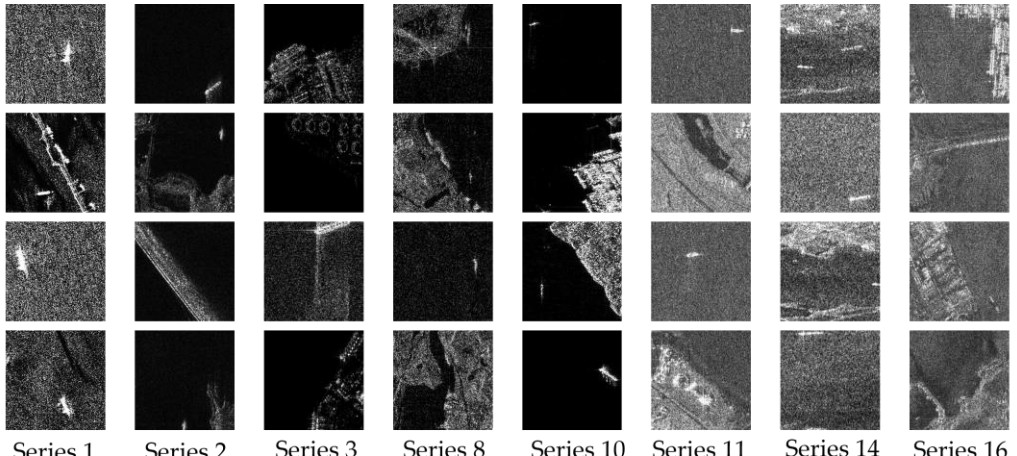

Series 1 Series 2 Series 3 Series 8 Series 10 Series 11 Series 14 Series 16

**Figure 8.** Some image samples from different series of the MSAR dataset.

There are two issues we would like to discuss in this manuscript:

(1) How the scaling of the training data affects the OOD generalization and the performance improvement of KGNN under different training sample sizes.
(2) How robust is the KGNN to independent factors concerning SAR data variation such as weather conditions, terrain type, and sensor characteristics.

To address the two issues above, two kinds of experiments were designed. Issue 1 is addressed with experiments under multiple-factor influence, and issue 2 is addressed with experiments under single-factor influence. The data descriptions of multiple-factor influence and single-factor influence are as follows:

(1) Multiple-Factor (MF) Influence Data Description:

Based on observations of the data series as shown in Figure 8, the series can be roughly divided into two kinds, one with a clean background and the other with a noisy background. For conventional CNN methods, if only samples with clean backgrounds are chosen as the training set, then most of the samples with noisy backgrounds will be misclassified as sea–land mixture scenes, and very little difference will be witnessed with the change in training sample size. Such a data configuration is not optimal for examining how variations in the size of the training samples affect OOD generalization. Therefore, we randomly select 10, 20, 50, 100, 200, and 500 samples only from series 1, 2, 12, and 13 to establish the training sets. We repeat the selection process five times to reduce the influence of

randomness in the training data selection. A total of 6374 samples were randomly selected from series 3~11 and 14~21 to establish the testing set. Samples with clean backgrounds and noisy backgrounds can both be found in the training and testing sets. However, the degree of background noise and overall layout of the samples differ between the two sets, influenced by a combination of factors such as weather conditions, terrain type, and sensor characteristics (MF influence). This data selection can effectively demonstrate how variations in the size of training samples affect the OOD generalization ability of all models.

(2) Single-Factor (SF) Influence Data Description:

Single factors (SFs) mean the independent factors that cause the SAR data variations, including weather conditions, terrain type, and sensor characteristics. The SFs in MSAR are as follows:

(a) SF-1 (weather conditions): weather conditions are related to the date information of the MSAR data. Here, we select data from series 7 and series 12 to establish the training and testing sets, respectively. As shown in Table 2, samples in series 7 are collected with the same satellite, same imaging mode, similar location (similar terrain type), but different weather conditions. A total of 50 samples from series 7 are selected as the training set, and 858 samples are selected from series 12 as the testing set.

**Table 2.** Data addressing weather condition variation in MSAR dataset, the yellow shading outlines the series we chose to establish the training set, the rest of the series were used to establish the testing set.

| Series | Image Index | Time | Location | Satellite | Imaging Mode |
|---|---|---|---|---|---|
| 7 | 8914~10,119 | 15 Jul. 2017 | E120.4, N35.4 | Gaofen-3 | FSII |
| 12 | 14,412~15,432 | 30 Sept. 2017 | E120.5, N36.3 | Gaofen-3 | FSII |

(b) SF-2 (terrain type): terrain type is related to the location information of the MSAR data. Here, we select data from series 3 to establish the training and testing sets. As shown in Table 3, samples 5252~5745 from series 3 are collected with the same satellite, imaging mode, and date, but quite different locations to samples 5937~8229 (about 670 km apart). A total of 50 samples from samples 5252~5745 are selected as the training set, and 1210 samples are selected from samples 5937~8229 as the testing set.

**Table 3.** Data addressing terrain type variation in MSAR dataset, the yellow shading outlines the series we chose to establish the training set, the rest of the series were used to establish the testing set.

| Series | Image Index | Time | Location | Satellite | Imaging Mode |
|---|---|---|---|---|---|
| 3 | 5252~5351 | 24 Oct. 2017 | E120.8, N36.1 | Gaofen-3 | FSI |
| | 5352~5745 | 24 Oct. 2017 | E120.9, N35.7 | Gaofen-3 | FSI |
| | 5937~8229 | 24 Oct. 2017 | E122.0, N30.1 | Gaofen-3 | FSI |

(c) SF-3 (sensor characteristics): there are no data collected with the same date and location but different imaging modes in the MSAR dataset. Here, we select data from series 2 and 13 and series 5, 8, and 16 to establish the training and testing sets, respectively. The main differences are their sensor characteristics (imaging modes), as shown in Table 4. A total of 50 samples from series 2 and 13 are selected as the training set, and 823 samples from series 5, 8, and 16 are selected as the testing set.

**Table 4.** Data addressing sensor characteristics variation in MSAR dataset, the yellow shading outlines the series we chose to establish the training set, the rest of the series were used to establish the testing set.

| Series | Image Index | Time | Location | Satellite | Imaging Mode |
|--------|-------------|------|----------|-----------|--------------|
| 2 | 613~5251 | 15 Jan. 2017 | E122.0, N30.3 | Gaofen-3 | FSI |
| 5 | 8244~8804 | 5 Jul. 2017 | E120.1, N35.8 | Gaofen-3 | QPSI |
| 8 | 10,120~12,972 | 3 Nov. 2017 | E121.9, N30.1 | Gaofen-3 | QPSI |
| 13 | 17,589~19,121 | 15 Feb. 2017 | E122.3, N29.9 | Gaofen-3 | FSI |
| 16 | 22,876~23,202 | 5 Oct. 2017 | E120.4, N36.2 | Gaofen-3 | QPSI |
| | 23,203~23,264 | 5 Oct. 2017 | E121.0, N33.5 | Gaofen-3 | QPSI |
| | 23,265~24,122 | 5 Oct. 2017 | E121.9, N30.1 | Gaofen-3 | QPSI |
| | 24,123~24,147 | 5 Oct. 2017 | E121.5, N34.2 | Gaofen-3 | QPSI |
| | 24,148~24,157 | 5 Oct. 2017 | E121.6, N34.7 | Gaofen-3 | QPSI |
| | 24,158~24,168 | 5 Oct. 2017 | E121.7, N35.0 | Gaofen-3 | QPSI |
| | 24,169~24,187 | 5 Oct. 2017 | E121.8, N35.6 | Gaofen-3 | QPSI |
| | 24,188~24,217 | 5 Oct. 2017 | E122.1, N36.7 | Gaofen-3 | QPSI |

*4.2. Experimental Setup*

We conducted extensive experiments to compare our proposed KGNN method with seven state-of-the-art classification models: ResNet-18 [48], ResNet-50 [48], GoogleNet [53], Inception-v3 [54], Xception [55], Efficient-b0 [56], and MobileNet-v2 [57]. These models are widely used for image classification tasks and have achieved remarkable performance on various benchmarks. The experiments were conducted on a workstation with CPU 12th intel i9 12900K, GPU NVIDA GeForce RTX 3090, and 32 GB RAM. We used Matlab 2021b for model implementation and data analysis. All networks were trained using the stochastic gradient descent with momentum (SGDM) algorithm. We set the mini-batch size at 32. The learning rate started from 0.01, and then multiplied by a drop factor of 0.9 every 10 epochs, and a momentum of 0.9 was employed. We removed the dropout layers in all models to achieve fitting to the training set more easily. All models were trained until the training accuracy stabilized at 100% for at least 20 epochs, which meant that the model had learned to fit the training data perfectly, while the total training epochs varied from 60 to 120. After training, all models were utilized on the testing set to make predictions. The test accuracy, which is the proportion of correct predictions to the total number of samples in the testing set, was employed to assess the KGNN's efficacy and comparative performance against the baseline methods.

*4.3. Results under Multiple-Factor Influence*

The number of trainable parameters and running time per image of each model are listed in Table 5. To evaluate the performance of different models, the mean values and standard deviations of the test accuracies are listed in Table 6. To view the difference between the performances of all models under all training sets in greater detail, we plot the mean values and standard deviations of the test accuracies with the change in training set sizes in Figure 9. Solid lines with an 'o' marker represent the average test accuracy, corresponding to the left-hand blue coordinate axis. Dashed lines with a '*' marker represent the standard deviation, corresponding to the right-hand orange coordinate axis. The performances of conventional data-driven deep learning models degenerate when there exists a distributional shift between the training and testing data. With 500 training samples, the best performance on the testing set of a conventional data-driven deep learning model comes from Inception-v3, which only reaches 94.86%, a relatively low index for a binary classification task with deep learning. The OOD performances of conventional data-driven deep learning models degenerate further as the size of the training set decreases.

**Table 5.** Model sizes and running time for all models.

| Model Name | Trainable Variables (Millions) | Running Time per Image (s) |
|---|---|---|
| KGNN | 14.4 | 0.0511 |
| ResNet-18 | 11.1 | 0.0013 |
| ResNet-50 | 23.5 | 0.0020 |
| GoogleNet | 5.9 | 0.0005 |
| Inception-v3 | 21.8 | 0.0013 |
| Xception | 20.8 | 0.0026 |
| Efficient-b0 | 4.0 | 0.0033 |
| MobileNet-v2 | 2.2 | 0.0018 |

**Table 6.** Classification performance obtained by different models with different training set sizes.

| Test Accuracy (%) | Number of Samples in Training Set | | | | | |
|---|---|---|---|---|---|---|
| | 10 | 20 | 50 | 100 | 200 | 500 |
| KGNN | 79.23 ± 3.29 | 87.07 ± 6.17 | 96.33 ± 0.91 | 97.41 ± 1.66 | 98.43 ± 0.27 | 98.75 ± 0.37 |
| ResNet-18 | 68.85 ± 7.53 | 70.12 ± 7.51 | 85.94 ± 2.04 | 89.60 ± 2.81 | 90.93 ± 1.52 | 91.61 ± 0.62 |
| ResNet-50 | 63.33 ± 5.58 | 72.99 ± 4.94 | 85.83± 2.07 | 88.29 ± 3.70 | 89.27 ± 1.39 | 91.22 ± 1.44 |
| GoogleNet | 65.31 ± 14.16 | 77.62 ± 4.04 | 84.67 ± 3.26 | 88.82 ± 2.76 | 88.95 ± 2.31 | 93.99 ± 0.96 |
| Inception-v3 | 57.50 ± 7.90 | 64.52 ± 11.10 | 84.13 ± 1.74 | 89.38 ± 2.04 | 92.67 ± 1.26 | 94.86 ± 2.25 |
| Xception | 62.41 ± 4.10 | 67.40 ± 6.01 | 83.39 ± 2.44 | 89.88 ± 2.12 | 90.79 ± 1.53 | 94.29 ± 1.19 |
| Efficient-b0 | 64.98 ± 2.12 | 65.72 ± 5.30 | 77.62 ± 3.82 | 83.58 ± 4.48 | 88.23 ± 2.72 | 91.26 ± 0.84 |
| MobileNet-v2 | 62.75 ± 5.24 | 70.67± 7.87 | 84.43 ± 1.46 | 84.72 ± 1.75 | 87.03 ± 1.61 | 89.48 ± 0.34 |

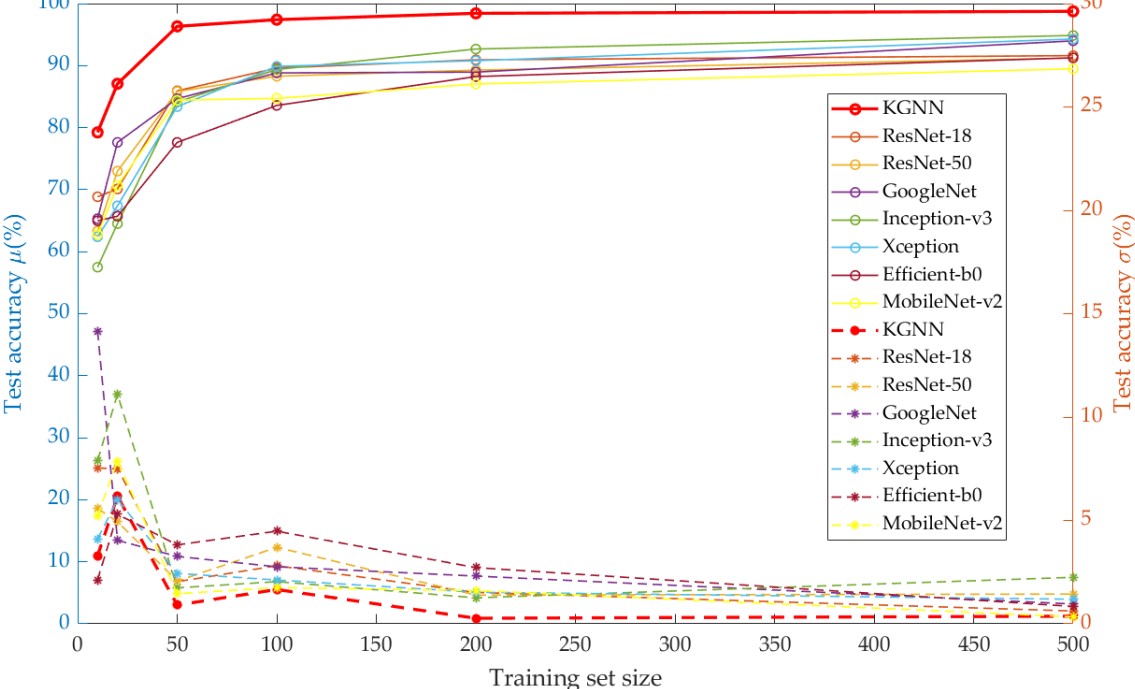

**Figure 9.** The average test accuracy and standard deviation of all models with different training set sizes. Solid lines with 'o' marker represent the average test accuracy, corresponding to the left-hand blue coordinate axis. Dash lines with '*' marker represent the standard deviation, corresponding to the right-hand orange coordinate axis.

However, the KGNN model outperformed all conventional deep learning models under all training set sizes. For a training set with 10 samples, the classification accuracy of our KGNN model reached 79.23%, approximately 15% higher than those of other

models. For a training set with 20 samples, the classification accuracy of our KGNN model reached 87.07%, approximately 17% higher than those of other models. Figure 9 shows the significant gap in the average test accuracy between our KGNN model and all other models. This gap generally narrows with the increase in training set size. However, the smallest gap value is still quite prominent (6%) with the training set size at 500.

The standard deviation of the test accuracy reflects the influence from the randomness of the training sample selection. As can be seen in Figure 9, the randomness affects the performance stability more seriously under small training set sizes, but in general, our KGNN method is less sensitive to this randomness than other models. The significant improvements in the average test accuracy between KGNN and other models, along with the resistance to the randomness of training sample selection, strongly demonstrate that integrating prior knowledge summarized by saliency analysis into a data-driven model can undoubtedly boost the latter's OOD generalization ability with limited training samples.

To demonstrate the improvement of the OOD generalization ability with limited training samples using the KGNN model more intuitively, we compare the classification results between KGNN and ResNet-18 from one experiment with a training set size of 50. Under identical experimental settings, the test accuracy of KGNN on the testing set is 96.30%, while that of ResNet-18 is 82.63%. There are 940 testing images that are correctly classified by KGNN but in the meantime misclassified by ResNet-18. Figure 10 shows several examples from the 940 images. From the first row of Figure 10, we can see that pure sea scenes with inhomogeneous sea clutter, a large target, a small target with energy leakage, multiple targets, and strong sea clutter are misclassified by ResNet-18. From the second row of Figure 10, we can see that sea–land mixture scenes with weak land clutter, a small proportion of land regions, unexpected land facilities like oil tanks, bright embankments, strong sea clutter, and confusing sea–land boundaries are misclassified by ResNet-18. However, the relations between the entities in these scenes are all generalized by the knowledge summarized from saliency analysis. As a result, through knowledge integration, the KGNN model can successfully classify all these challenging scenarios with a training set that is very limited in both diversity and quantity.

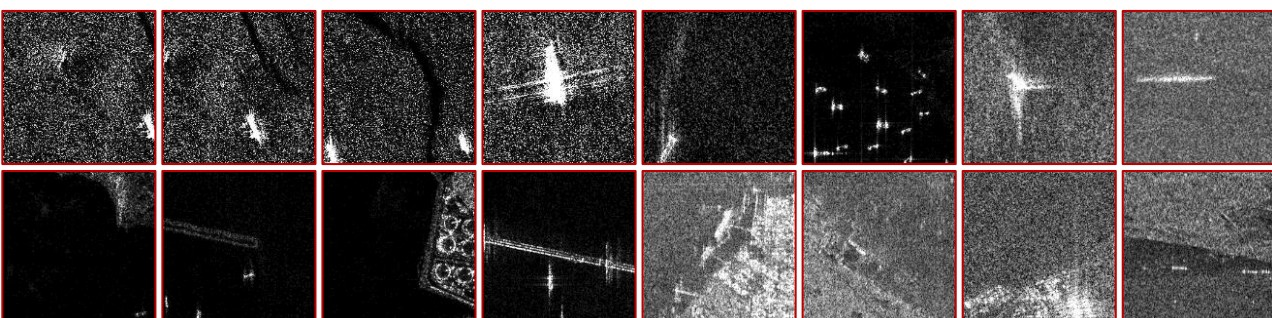

**Figure 10.** Examples chosen from 940 testing images that are correctly classified by KGNN but misclassified by ResNet-18.

Figure 11 shows the training process of all models with the first set of 200 and 500 training samples. For the KGNN model, it generally takes less than 10 epochs to stabilize at 100% training accuracy, while for other models it requires at least 60 epochs to reach the same training performance. This difference indicates that it is much easier to find the global minimum in the hyperparameter space with the KGNN model. The guidance of prior knowledge can greatly facilitate the training process of deep neural networks.

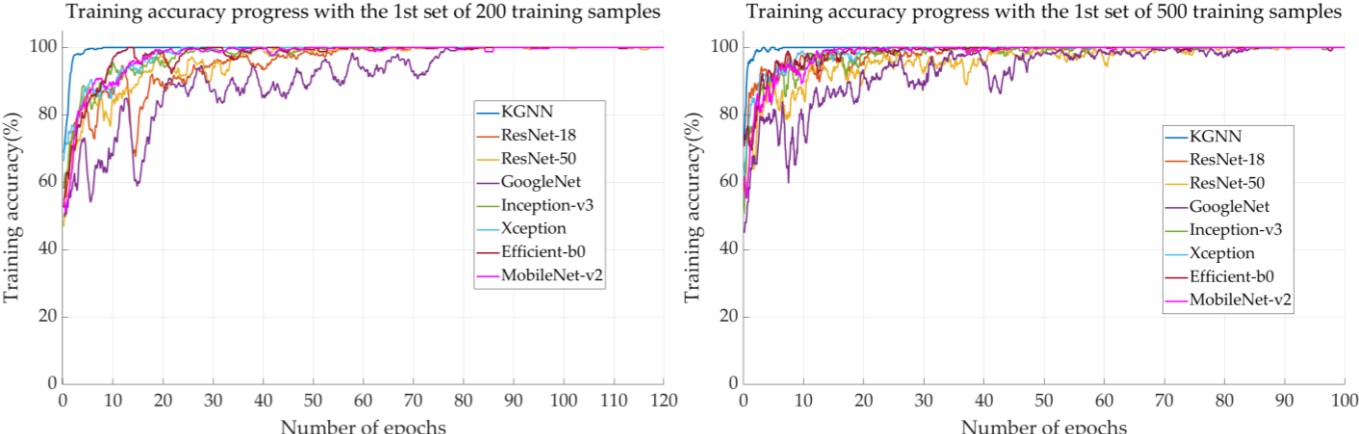

**Figure 11.** Training process of all models with the first set of 200 and 500 training samples.

*4.4. Results under Single-Factor Influence*

The results under multiple-factor influence give a comprehensive assessment of KGNN's OOD generalization ability across varying sizes of training sets. Following this, we will analyze the OOD generalization ability under the influence of different independent factors.

(1) Results under SF-1 (weather condition): detailed information about the training and testing sets is listed in Table 2. Some samples from the training and testing sets are shown in Figure 12. There is a significant difference in the overall intensity of images under the influence of different weather conditions. Figure 13 shows the test accuracy of all models, the KGNN method shows a significant improvement in OOD generalization ability (approximately 70% higher than other models). The reason for the poor performance of conventional CNN methods is that they generally identify all test samples as sea–land mixture scenes due to the image intensity variation under different weather conditions.

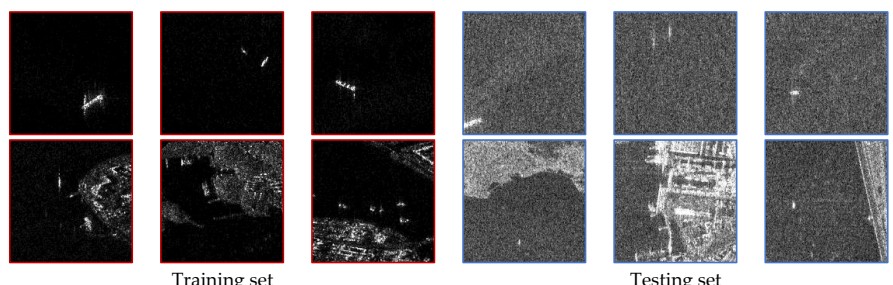
Training set        Testing set

**Figure 12.** Samples from training and testing sets in SF-1.

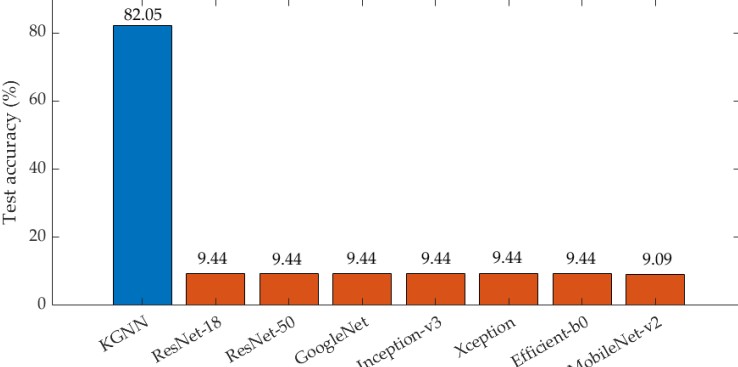

**Figure 13.** Test accuracy of all models under SF-1.

(2) Results under SF-2 (terrain type): detailed information about the training and testing sets is listed in Table 3. Some samples from the training and testing sets are shown in Figure 14. For pure sea scenes, the main variation lies in the target type. For sea–land mixture scenes, the training set mainly consists of natural islands, while the testing set is mainly composed of oil tanks and dock facilities. Figure 15 shows the test accuracy of all models, the KGNN method shows superior OOD generalization ability compared to other models with an approximately 15% improvement in the test accuracy. The KGNN model is more robust in dealing with distributional shifts caused by terrain or target types.

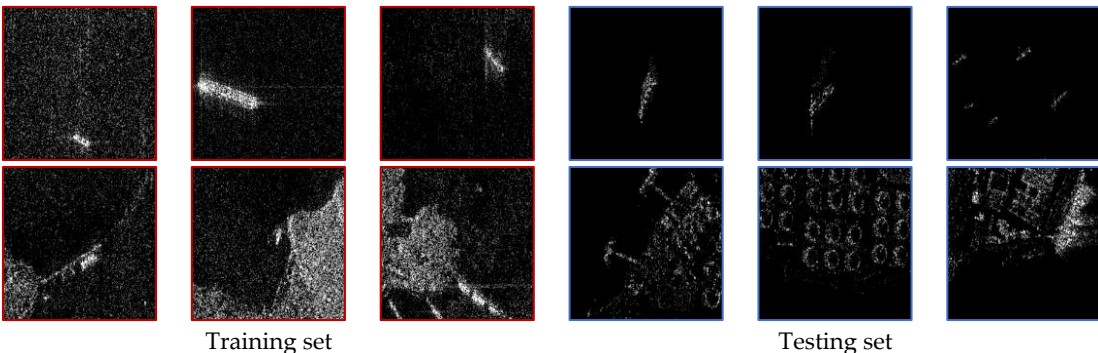

Training set                  Testing set

**Figure 14.** Samples from training and testing sets in SF-2.

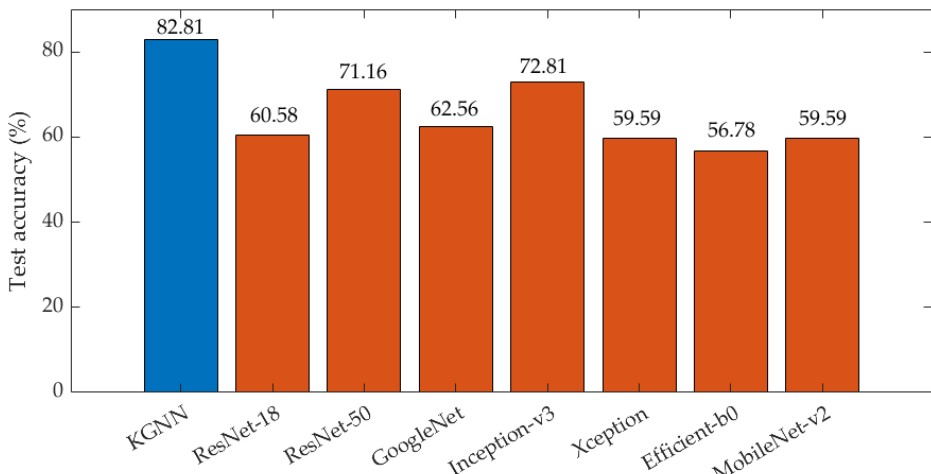

**Figure 15.** Test accuracy of all models under SF-2.

(3) Results under SF-3 (sensor characteristics): detailed information about the training and testing sets is listed in Table 4. Some samples from the training and testing sets are shown in Figure 16. From observations, the distributional shifts between the training and testing sets are not as prominent as those in SF-1 or SF-2. Figure 17 shows the test accuracy of all the models, the test accuracy of the KGNN method is approximately 2%~12% higher than other models, indicating a higher robustness in dealing with sensor characteristic variations.

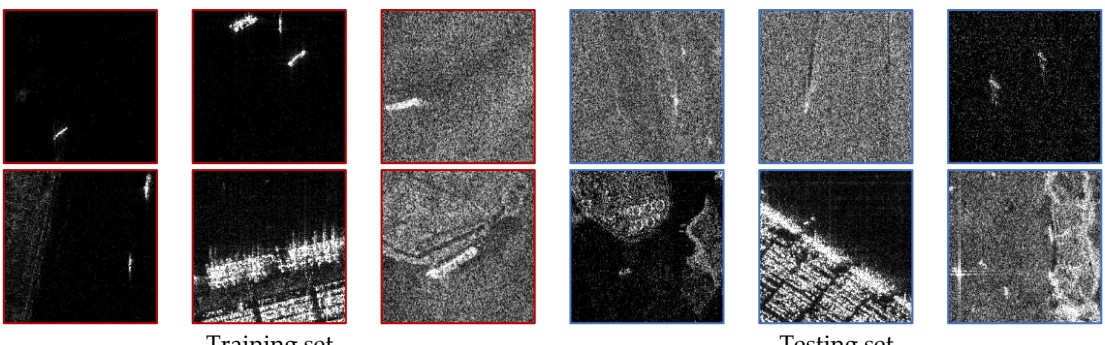

Training set                    Testing set

**Figure 16.** Samples from training and testing sets in SF-3.

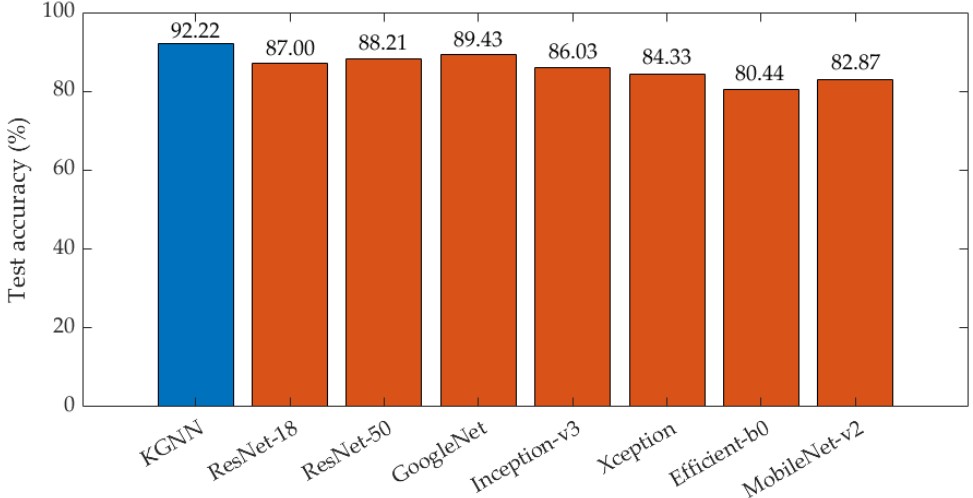

**Figure 17.** Test accuracy of all models under SF-3.

### 4.5. Discussion

To fully demonstrate the OOD generalization ability of the proposed KGNN model with limited training samples, two kinds of experiments were conducted. For experiments under MF influence, the distributional shift between the training and testing sets is determined by the comprehensive influence of weather conditions, terrain type, and sensor characteristics. The results under MF influence show that the OOD generalization abilities of all the models degenerate with a decrease in the training sample size, while the KGNN model outperformed all the conventional data-driven deep learning models under all training set sizes, with a significant test accuracy increase from about 6% to 17%. From the standard deviation of the test accuracy, KGNN is less sensitive to sample selection randomness than other models. Image comparisons show that the KGNN model improves the OOD generalization with a limited training set, especially on pure sea scene samples with inhomogeneous sea clutter, a large target, a small target with energy leakage, multiple targets, and strong sea clutter, and on sea–land mixture scene samples with weak land clutter, a small proportion of land regions, unexpected land facilities like oil tanks, bright embankment, strong sea clutter, and confusing sea–land boundaries. Moreover, the integration of knowledge can prominently facilitate the training process. The KGNN model converges within many fewer epochs than conventional data-driven deep learning models.

For experiments under SF influence, the distributional shift between the training and testing sets is determined by the independent influence of weather conditions, terrain type, and sensor characteristics. The results under SF influence show that the greatest distributional shift between the training and testing sets is caused by weather conditions, followed by terrain and target type variations. The least impact comes from sensor characteristics.

The OOD generalization ability of KGNN is approximately 70% higher than other models under the influence of different weather conditions. Under terrain and target influences, the improvement is around 15%, and under the influence of sensor characteristics, the improvement ranges from 2% to 12%.

The main drawback of the KGNN model is the processing time. However, for SAR scene classification, this delay will not cause problems. To accommodate continuous imaging, it is generally required to ensure that the program processing time does not exceed the time it takes to output a single frame. SAR imaging needs time to accumulate echo pulse signals during the synthetic-aperture interval, followed by post-processing to convert these signals into an SAR image [58]. It usually takes a few seconds to several minutes to generate an SAR image depending on the resolution, coverage, and algorithm [58]. Therefore, despite the processing time being longer compared to other models, the KGNN remains sufficiently rapid for SAR perception tasks.

## 5. Conclusions

This paper proposed a novel knowledge-guided neural network (KGNN) model by integrating knowledge summarized by saliency analysis for maritime SAR image classification to improve OOD generalization performance with limited training data. Knowledge reflecting the inherent relations between entities in various SAR image scenes is summarized via saliency analysis. A knowledge integration strategy is then designed to incorporate the descriptive knowledge into a ResNet-18 backbone. Specifically, all saliency maps along with the original image are input into the first several convolutional and residual blocks of ResNet-18 separately to generate individual deep feature embeddings, then all feature embeddings are concatenated and propagate together into the rest of the residual blocks of ResNet-18 to generate scene type classification results. The experimental results demonstrate that the pre-identification results with the saliency map in this paper can be a boost and supplement for the data-driven CNN-based model in OOD generalization, although their performance may deteriorate in some complex scenes. The information of the pre-identification results and their inherent relations can both be addressed by the feature integration strategy. From the improvements, it is inferred that the deterioration part in the saliency maps is corrected by their inherent relations. In addition, the KGNN model shows robustness in OOD scenarios caused by weather conditions, terrain type, and sensor characteristics. In summary, the proposed KGNN model provides a way to integrate knowledge summarized by saliency analysis of maritime SAR scene classification into the data-driven model, boosting the latter's OOD generalization ability under limited training samples, which is very important for practical applications with special SAR imaging platforms.

In future work, we will consider reducing the deterioration part in the saliency maps to further boost the OOD generalization ability of the KGNN model and applying the idea of this method in OOD semantic segmentation tasks. We will also consider applying this method to other detection systems including optical and infrared modalities.

**Author Contributions:** Conceptualization, Z.C. and X.Z. (Xiaoling Zhang); methodology, Z.C. and Z.D.; software, Z.C.; validation, Z.C. and Xiaoling Zhang.; formal analysis, Z.C.; investigation, Z.C.; resources, Z.C. and X.Z. (Xin Zhang); data curation, Z.C.; writing—original draft preparation, Z.C.; writing—review and editing, Z.C. and X.Z. (Xiaoling Zhang); visualization, Z.C.; supervision, Z.D. and T.Q.; project administration, Z.D. and T.Q.; funding acquisition, Z.D. and X.Z. (Xin Zhang). All authors have read and agreed to the published version of the manuscript.

**Funding:** This research received no external funding.

**Data Availability Statement:** Data sharing is not applicable to this article. The MSAR dataset is available from https://radars.ac.cn/web/data/getData?dataType=MSAR (accessed on 8 August 2023).

**Acknowledgments:** The authors would like to thank the editors and anonymous reviewers for their valuable comments that greatly improved our manuscript.

**Conflicts of Interest:** Author Zhe Chen, Zhiquan Ding, Xin Zhang, and Tianqi Qin were employed by the company Multisensor Intelligent Detection and Recognition Technologies R&D Center of CASC. The remaining author declares that the research was conducted in the absence of any commercial or financial relationships that could be construed as a potential conflict of interest.

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
