# Peer review of "Improving Out-of-Distribution Generalization in SAR Image Scene Classification with Limited Training Samples"

_remotesensing, doi:10.3390/rs15245761_

Round 1

Reviewer 1 Report

Comments and Suggestions for Authors

The paper introduces a knowledge-guided neural network (KGNN) model for maritime SAR scenes, showing significant accuracy and improved OOD generalization ability on unseen complex scenes.

 The Research paper provides respectable findings and is well-written, but before it is accepted, it needs to be strengthened in the following ways:

 The introduction could be expanded, and more related research sources should be cited. The author may use the following sources: https://doi.org/10.3390/electronics12061421 and DOI: 10.1109/ICCCIS48478.2019.8974502  

 Discuss the principles and advantages of Improving Out-of-distribution Generalization in SAR Image Scene Classification with Limited Training Samples.

 The author should  explain how the proposed techniques enhance generalization in such scenarios, and specific improvements observed.

 what are the potential challenges or limitations associated with implementing this approach in real-world scenarios? Provide examples or case studies to support your discussion.

 The author should discuss, Which performance criteria were employed to assess the suggested techniques' efficacy and their comparative performance against the baseline methods.

 The author should discuss the  robustness of the proposed techniques in handling variations in SAR data, such as weather conditions, terrain types, and sensor characteristics.

The author should mention the name of the software/tools used for data analysis.

The author must include logical justifications for the results, limitations, and recommendations for future performance enhancement studies. 

Comments on the Quality of English Language

Please try to write more simply so that your paper will be understood by all readers.

The paper requires careful polishing of its English presentation, addressing grammatical issues,  and addressing typos and poorly written sentences.  

Reviewer 2 Report

Comments and Suggestions for Authors

The manuscript:

“Improving Out-of-distribution Generalization in SAR Image Scene Classification with Limited Training Samples”, by Z. Chen, Z. Ding, X. Zhang, X. Zhang and T. Qin (Ref. No.: remotesensing-2703283-peer-review-v1),

contains some interesting material. However, it requires a significant elaboration. In particular, the objective and novelty of this work should be stronger emphasized. Furthermore, the Table 2 shows that the proposed model KNGG is slowest by the order of the magnitude (running time per image) as compared to other existing methods. It is not clear why KNGG model is justifiable for practical applications? Furthermore, the authors considered only the advantages of the proposed model and did not discuss about its disadvantages.

English is acceptable. However, minor corrections may be advisable. Moreover, the manuscript requires a few citations.

Apart from this the following should be taken into consideration:

Abstract

1) The size of the Abstract is somehow large and should be shortened. It is enough to show the key results obtained in this study.

1. Introduction

1) What is the objective and novelty (or originality) of this work? These two questions should be briefly described at the end of the Introduction.

2) Briefly describe the achievements and advantages of the proposed design/method.

2. Related Works

1) This section greatly overlaps with Introduction since both sections provide brief survey. Perhaps these two sections should be merged.

3. Materials and Methods

1) The sentence: “In this work, we assume knowledge as task-specific information about relations between entities in maritime SAR image scene. We extract the knowledge in descriptive sentence form through saliency analysis in Section 3.1, and propose a knowledge guided neural network (KGNN) to incorporate the knowledge in a deep learning backbone in Section 3.2”. Is this method new? If yes, the novelty should be emphasized.

2) Sources for the equations (1) and (2) should be cited.

3) The sentence: “However, for inhomogeneous scenes, detection methods using a global intensity threshold would not be suitable since many false alarms such as strong sea clutters, noises, land facilities, land clutters will also be detected”, should be cited.

4) Equation (3) should be cited.

5) The sentence: “For SM3, there are entities as bridges and land clutters misidentified at the edge of land regions”, is not clear and should be clarified.

4. Results

1) The sentence: “We randomly select 10, 20, 50, 100, 200, and 500 samples only from series 1, 2, 13, 14”. What are criteria for these specific selections?

2) Table 2. Why KGNN model takes significantly more time per page? Does this delay cause the problems?

5. Discussion and conclusions

1) Perhaps the discussion should be moved to the section Results and the section Results should be renamed as Results and Discussion. It is not conventional and inconvenient to mix up discussion and conclusions.

2) The sentence: “Concretely, all saliency maps along with the original image …”. Should be rewritten as “Specifically, all saliency maps along with the original image …”. This will improve English.

3) The sentence: “All models were trained until a stable 100% training accuracy was achieved …”. It cannot be 100% training accuracy. May be “close to 100%” is more correct?

4) What are the drawbacks of the KGNN model?

5) Table 2 shows that KGNN model provides running time per image 0.0511 s, which is by order of the magnitude slower than the other existing models. However, this is a sentence stating that: “… the KGNN model converges much faster than conventional data-driven deep learning models in the training process”. It is a contradiction between slowest running time per image and this claim. This issue should be clarified.

The manuscript requires a major mandatory revision.

Comments on the Quality of English Language

English is acceptable.

Reviewer 3 Report

Comments and Suggestions for Authors

The abstract is too long. The issue is not well written. Much material concerning the state of the art is typecast, which should either be deleted or transferred to the introduction. The abstract should be shortened considerably and divided into the following parts: The problem of..... is......... In this papaer we........ Results are........ Performance are.......

The analytical model should be better described. An example Supervised Model Learning is not adequately described, but neither is the model belonging to the proposed method. E.g. the "Knowledge Extraction by Saliency Analysis" method should be better described.

The vormulae should be better explained. Besides the analytical models need to be better described, correct notation must be used. Then a strategy is required to distinguish different parameters such as scalars, vectors and matrices, and also tensors.

I do not understand whether SAR images are processed within the complete domain of existence, thus processing the entirety of the complex data, or only with energy, thus processing only the modulus contribution.

The experimental results are not clear. must better demonstrate the functionality of the proposed algorithm

Comments on the Quality of English Language

No issues found

Round 2

Reviewer 3 Report

Comments and Suggestions for Authors

Accepted

Comments on the Quality of English Language

The qualit is good